# Investigating Practical Impacts of Using Single-Antenna and Dual-Antenna GNSS/INS Sensors in UAS-Lidar Applications

**DOI:** 10.3390/s21165382

**Published:** 2021-08-09

**Authors:** Ryan G. Brazeal, Benjamin E. Wilkinson, Adam R. Benjamin

**Affiliations:** 1Geomatics Program, School of Forest, Fisheries, and Geomatics Sciences, University of Florida, Gainesville, FL 32611, USA; ryan.brazeal@ufl.edu; 2Geospatial Modeling and Applications Lab, School of Forest, Fisheries, and Geomatics Sciences, University of Florida, Gainesville, FL 32611, USA; 3Geomatics Program, Fort Lauderdale Research & Education Center, School of Forest, Fisheries, and Geomatics Sciences, University of Florida, Fort Lauderdale, FL 33314, USA; abenjamin1@ufl.edu

**Keywords:** unoccupied aerial system, UAV, drone, GPS, inertial, heading, mobile mapping

## Abstract

Data collected from a moving lidar sensor can produce an accurate digital representation of the physical environment that is scanned, provided the time-dependent positions and orientations of the lidar sensor can be determined. The most widely used approach to determining these positions and orientations is to collect data with a GNSS/INS sensor. The use of dual-antenna GNSS/INS sensors within commercial UAS-lidar systems is uncommon due to the higher cost and more complex installation of the GNSS antennas. This study investigates the impacts of using a single-antenna and dual-antenna GNSS/INS MEMS-based sensor on the positional precision of a UAS-lidar generated point cloud, with an emphasis on the different heading determination techniques employed by each type of GNSS/INS sensor. Specifically, the impacts that sensor velocity and acceleration (single-antenna), and a GNSS compass (dual-antenna) have on heading precision are investigated. Results indicate that at the slower flying speeds often used by UAS (≤5 m/s), a dual-antenna GNSS/INS sensor can improve heading precision by up to a factor of five relative to a single-antenna GNSS/INS sensor, and that a point of diminishing returns for the improvement of heading precision exists at a flying speed of approximately 15 m/s for single-antenna GNSS/INS sensors. Additionally, a simple estimator for the expected heading precision of a single-antenna GNSS/INS sensor based on flying speed is presented. Utilizing UAS-lidar mapping systems with dual-antenna GNSS/INS sensors provides reliable, robust, and higher precision heading estimates, resulting in point clouds with higher accuracy and precision.

## 1. Introduction

Three-dimensional mapping of natural or built environments using high-resolution light detection and ranging (lidar) sensors onboard unoccupied aerial systems (UAS) is now a common practice. Recent advancements in robotics and real-time computing have led to the possibility of autonomous UAS-lidar mapping operations in global navigation satellite system (GNSS)-denied environments (e.g., underground mines) [1]. However, the more common application of autonomous UAS-lidar mapping still occurs in outdoor environments [2]. For observations collected from a moving lidar sensor to produce an accurate digital representation (i.e., point cloud) of the physical environment that is scanned, the time-dependent positions and orientations of the lidar sensor with respect to a fixed reference frame are needed. The most widely used approach to determining these positions and orientations is to collect data with a GNSS-aided inertial navigation system (GNSS/INS) [3]. As a result of advancements in the miniaturization of electronics, GNSS/INS sensors containing high-accuracy GNSS receivers and micro-electro-mechanical system (MEMS)-based inertial measurement units (IMUs) are now commercially available for UAS applications [4,5]. Using direct georeferencing (DG), lidar observations are fused with GNSS/INS-derived positions and orientations based on a common time reference to produce a georeferenced point cloud [6]. To investigate the positional precision of a point cloud produced by a DG UAS-lidar mapping system, the positioning and navigation aspects of GNSS and INS must be examined.

A GNSS receiver can estimate position, velocity, and time (PVT) by observing encoded radio frequency (RF) signals from a constellation of medium Earth-orbit satellites. The encoded RF signal from each satellite allows a GNSS receiver to estimate the signal’s travel (transit) time, the satellite’s clock error, and the position of the satellite when the signal was transmitted within a global reference frame. The transit time estimates are multiplied by the speed of light to produce range estimates between the GNSS receiver and the satellites. However, the transit time estimates are all biased by a common systematic error known as the receiver’s clock offset. By observing encoded RF signals from a minimum of four satellites, a GNSS receiver can simultaneously estimate its clock offset to a precision of less than 30 nanoseconds [7] and its three-dimensional (3D) position within a global reference frame. The encoded RF signals also allow a GNSS receiver to measure the respective Doppler frequency shift for each observed signal. The Doppler frequency shifts of the received signals are produced by the user-satellite relative motion and enable velocity estimates with a precision of a few centimeters per second to be realized [8]. If time-differenced carrier phase (TDCP) methods are used, the precision of the velocity estimates can improve to a few millimeters per second [ibid.]. The GNSS-based PVT estimates have high long-term accuracy, have bounded errors (i.e., no drift), and are independent of previous estimates.

An INS consists of triads of orthogonally mounted accelerometers and gyroscopes, collectively called an IMU, in addition to mechanization algorithms, a gravity model, and a computer [9]. The accelerometers are used to measure the specific forces (i.e., non-gravitational accelerations) experienced relative to the orthogonal sensing directions. Integrating the acceleration measurements with respect to time produces velocity estimates and a second integration with respect to time produces position estimates within a local reference frame. The gyroscopes measure the angular rates regarding the orthogonal sensing directions. Integrating the angular rate measurements with respect to time produces angular orientation estimates for the sensing directions within a local frame. Starting with a gravity model and the initial values for position, velocity, and orientation, the computer operates on the IMU’s outputs to furnish the current position, velocity, and orientation [9]; thus, an INS is a sophisticated implementation of dead reckoning [10]. The INS-based position, velocity, and orientation estimates have high short-term accuracy, have unbounded errors (i.e., drift due to accumulation of errors), and are dependent on previous estimates.

Capitalizing on the complementary characteristics of GNSS and INS, their synergistic integration overcomes their individual drawbacks and provides a more accurate and robust navigation solution than either could achieve on its own [9]. Estimation techniques, predominantly based on Kalman filtering, are employed to optimally fuse the GNSS and IMU observations to yield a reliable navigation solution [ibid]. The optimization techniques are classified as either loosely coupled, tightly coupled, or ultra-tightly coupled, and indicate how and what information from each system is used to improve the accuracy, precision, and robustness of the fused navigation solution. Each technique has an associated level of complexity, achievable accuracy and precision, and application suitability. Table 1 summarizes the important and complimentary characteristics of GNSS and INS.

The navigation solution—commonly referred to as a trajectory—contains the time-tagged position estimates (e.g., latitude, longitude, and ellipsoidal height), angular orientation estimates (e.g., roll, pitch, and heading), and velocity estimates for the GNSS/INS sensor. These estimates are commonly reported at rates of 200 Hz or higher. The highest accuracy position estimates are achieved when differential GNSS techniques based on the carrier phase observable are used, provided the carrier phase ambiguity terms are resolved to their correct integer values. High accuracy roll and pitch orientation estimates are achieved when the direction of gravity is precisely determined by the accelerometers, which is improved when GNSS measurements are used to account for the impact of dynamic motion on the accelerometer readings [11]. Unlike the roll and pitch estimates, which are measured with respect to the direction of gravity, the heading orientation estimate is measured with respect to the geographic (true) north direction. Common approaches used by GNSS/INS sensors for determining the true north direction include using additional hardware sensors (e.g., a magnetometer or a GNSS compass) and data-driven methods (e.g., GNSS course over ground, gyro-compassing, and dynamic alignment) [11,12,13].

For single-antenna GNSS/INS sensors, a combination of a triaxial magnetometer and dynamic alignment is widely employed for heading determination. This type of GNSS/INS sensor loses the observability of heading during low-dynamic or static situations in which dynamic alignment becomes impossible and commonly falls back on the lower accuracy magnetometers’ observations to continue stabilizing heading [11]. For dual-antenna GNSS/INS sensors, a combination of a GNSS compass and dynamic alignment are commonly used for heading determination. A GNSS compass can determine the relative displacement (i.e., vector) between two GNSS antennas to millimeter-level precision in real-time using carrier phase measurements and does not require sensor motion as is the case for dynamic alignment [11]. However, the fixed distance between the GNSS antennas—referred to as the compass baseline length—affects the achievable precision of the heading estimate. As the distance between GNSS compass antennas is decreased, with short distances often unavoidable onboard a UAS (<1 m typical), the heading precision improvement provided by the GNSS compass will diminish. In practice, dual-antenna GNSS/INS sensors rely on dynamic alignment when available but during periods of low-dynamics, the GNSS compass solution is used instead [11]. Although single and dual-antenna GNSS/INS sensors both utilize dynamic alignment, dual-antenna systems almost always provide more precise heading estimates, along with increased performance and robustness. However, the use of dual-antenna GNSS/INS sensors within commercial UAS-lidar systems is fairly uncommon due to the increased system complexity, added weight, higher cost, and additional calibration requirements. Thus, the focus of this study is to understand the impacts of using a single-antenna versus a dual-antenna GNSS/INS MEMS-based sensor on the positional precision of a UAS-lidar generated point cloud and to illustrate the potential practical benefits of a dual-antenna system to contrast with the potential drawbacks.

Only a few studies have been published that partially address the impact of GNSS/INS heading estimates on the positional precision of UAS-lidar point clouds. In [14], a theoretical analysis was performed that investigated and quantified the effects of trajectory errors, lidar observation errors, and beam incidence angles on point cloud coordinate accuracies. The introduced error values and flight parameters (e.g., flying height) were typical of those found within commercial UAS-lidar systems and were experienced during mapping operations. The study concluded that error in the yaw (heading) angle had the greatest impact on the horizontal accuracy of a point cloud. In [15], an empirical accuracy assessment of a typical UAS-lidar system utilizing a single-antenna GNSS/INS sensor was presented. A UAS-lidar point cloud collected during a low-altitude/low-speed flight was compared to a reference point cloud generated by a terrestrial laser scanner (TLS). An analysis of the GNSS/INS orientation estimates showed lower than expected accuracy in the azimuth (heading) estimates and was attributed to poor kinematic alignment of the initial orientation. As a result, the accuracy of the UAS-lidar point cloud was decreased. The study concluded that absolute accuracy of the UAS-lidar point cloud was no worse than 10 cm and suggested the precision was as good as 2 cm. In [14] and [15], the theoretical and experimental findings indicated that the precision of the heading orientation estimates had the largest impact on the positional precision of a UAS-lidar point cloud. To build upon these findings, this study conducted a controlled UAS-lidar mapping experiment to examine how the precision of heading orientation estimates from single and dual-antenna systems affects the positional precision of a resulting point cloud. Specifically, the objectives of this study are to:provide the theoretical foundations behind different heading determination techniques used by GNSS/INS sensors;detail the practical considerations for implementing these heading determination techniques during UAS-lidar field operations (e.g., INS initialization procedures and mission planning considerations);compare the precisions of heading orientation estimates reported by a single-antenna and a dual-antenna GNSS/INS sensor at different UAS flying speeds; andassess the positional precisions of UAS-lidar point clouds generated from a single-antenna and a dual-antenna GNSS/INS sensor.

## 2. Materials and Methods

### 2.1. Equipment/Sensor

An Applanix APX-18 UAV sensor (Applanix, Richmond Hill, Canada) was used to collect the GNSS/INS data for this study (Figure 1). The APX-18 is a commercially available small form factor dual-antenna GNSS/INS sensor specifically designed for UAS mapping applications. It is an original equipment manufacturer (OEM) GNSS/INS solution designed to georeference lidar and other imaging data when collected from UAS at low speeds or when hovering [16]. The dimensions of the APX-18 board are 100 mm by 60 mm and by 12 mm, and weighs 62 g. Observations from the dual GNSS antennas on the APX-18 enables the use of Applanix’s GNSS Azimuth Measurement Subsystem (GAMS). GAMS is essentially a GNSS compass that is integrated into a tightly coupled Kalman filter in which the GNSS heading estimates are blended with the heading estimates produced from the dynamic alignment of the INS [13,17]. GAMS provides the ability to compute precise heading estimates independent of latitude and vehicle dynamics [17].

### 2.2. GNSS/INS Heading Determination Techniques

The use of a magnetometer for heading determination is the simplest method as the magnetic field of the Earth can be directly measured and used to infer the direction of true north. However, the magnetic vector at the position of the magnetometer will likely have contributions from other sources (e.g., naturally occurring magnetic material in the ground and ferrous materials from the attached vehicle) [11]. Therefore, proper calibration of the magnetometer is required to achieve highest accuracy measurements [18]. Under ideal conditions (e.g., stationary within an open area), a magnetometer can produce heading estimates with an accuracy of <0.5°, which is suitable for initializing the heading within an INS [19].

Most GNSS/INS sensors used in commercial UAS-lidar systems including the Applanix APX-18 UAV utilize MEMS-based IMUs. These types of IMUs often experience gyroscope bias and noise errors that exceed the Earth’s rotation rate (i.e., >15°/hour); therefore, the stationary gyro-compassing method for determining heading is not feasible [19]. The precision of the heading estimate using the gyro-compassing method is inversely proportional to cosine of latitude, as expressed in Equation (1), and will therefore decrease at higher latitudes [11].

(1)$$\sigma_{\psi }\propto \frac{1}{\cos\left(\varphi \right)}$$
where $\sigma_{\psi }$ is the precision of the heading estimate and φ is the geodetic latitude.

The primary benefit of a dual-antenna GNSS/INS sensor is the capability of using the GNSS compass method for heading determination. A GNSS compass is able to compute the heading angle estimate, measured as the clockwise angle from true North, and the elevation (tilt) angle estimate of the compass vector within a local level (east–north–up) frame (Figure 2). The precision of the tilt angle estimate is dependent on the precision of the vertical component of a GNSS compass vector, which for GNSS positioning is commonly 2–3 times less precise than the horizontal components [20]. As a result, the tilt angle estimates provided by a GNSS compass are not commonly used within the GNSS/INS navigation filter [11]. However, [20] demonstrated that if both receivers used in a GNSS compass utilize a common clock, the precision of the vertical component could be close to those of the horizontal components for a GNSS compass vector. When using a GNSS compass, a few added challenges that must be overcome include maintaining a direct line-of sight to the satellites, observing at least six common satellites between the two antennas, and reducing multipath interference [11].

The GNSS compass method employs differential GNSS positioning using carrier phase measurements to precisely determine the relative displacements of the GNSS compass vector within a local frame. For carrier phase measurements to produce precise results, the unknown ambiguity terms must be correctly resolved to their integer values. Once this has been done successfully, the carrier phase measurements will act as very precise pseudo-range data, which makes very precise orientation determination possible [20]. Using this method, the differential position of the secondary GNSS antenna with respect to the primary GNSS antenna will have a precision of between 2 and 10 mm, depending on the local multipath environment [13]. Following the GNSS compass error propagation presented in [20], the precision of the heading estimate is expressed by:
(2)$$\sigma_{\psi }=\sqrt{\frac{{\left(\cos\;\psi \right)}^{2}\;\sigma_{e}{}^{2}+{\left(\sin\;\psi \right)}^{2}\;\sigma_{n}{}^{2}}{L^{2}{\left(\cos\;\theta \right)}^{2}}}$$
where $\sigma_{e}$ and $\sigma_{n}$ are the precisions of the relative displacements in the east and north directions, respectively, between the GNSS antennas as determined by differential carrier phase positioning, and L is the fixed length of the compass baseline. If it is assumed that the precisions of the relative horizontal displacements are equal (i.e., $\sigma_{e}=\sigma_{n}$) and the GNSS compass is operated while nominally level (i.e., θ≈0°), Equation (2) simplifies and is now expressed by:
(3)$$\sigma_{\psi }=\frac{\sigma_{h}}{L}$$
where *σ_h_* is the precision of the relative displacement between the GNSS antennas in the local horizontal plane. Figure 3 illustrates the theoretical heading precisions of the GNSS compass method for different compass baseline lengths.

Both single-antenna and dual-antenna GNSS/INS sensors utilize the dynamic alignment method for heading determination during UAS-lidar mapping operations. Dynamic heading alignment works by comparing the accelerometer measurements from the INS solution and the position and velocity measurements from the GNSS solution to estimate the internal dynamics of the system [11,13]. The method works best when the sensor experiences rapid changes of direction and varying forces of acceleration [13]. The complex computations utilized by a GNSS/INS navigation filter to estimate heading and its associated precision are beyond the scope of this study; see [9] for further information.

A simpler concept that illustrates how sensor dynamics can be used to determine heading is the GNSS velocity method, which is sometimes used to initialize heading within the dynamic alignment method [19]. The GNSS velocity method uses a trigonometric relationship based on GNSS-derived velocity estimates within a local reference frame to determine heading. It is expressed by:
(4)$$\psi =\mathrm{atan}\left(\frac{v_{e}}{v_{n}}\right)+c$$
where $v_{e}$ and $v_{n}$ are the east and north components of the GNSS-derived velocity, respectively, and c is the misalignment angle between the GNSS velocity vector and the forward axis of the navigation reference frame. The precision of an optimistic heading estimate can be calculated using uncorrelated variance propagation and is expressed by:
(5)$$\sigma_{\psi }=\sqrt{\frac{v_{n}{}^{2}}{{\left(v_{n}{}^{2}+v_{e}{}^{2}\right)}^{2}}\cdot \sigma_{v_{e}}{}^{2}+\frac{v_{e}{}^{2}}{{\left(v_{n}{}^{2}+v_{e}{}^{2}\right)}^{2}}\cdot \sigma_{v_{n}}{}^{2}+\sigma_{c}{}^{2}}$$
where $\sigma_{v_{e}}$ and $\sigma_{v_{n}}$ are the precisions of the GNSS-derived velocity components in the east and north directions, respectively, and $\sigma_{c}$ is the precision in the misalignment angle that represents the achievable limit for the heading precision (i.e., best case) using the GNSS velocity method. If it is assumed that the precisions of the horizontal velocity components are equal (i.e., $\sigma_{v_{e}}=\sigma_{v_{n}}$), Equation (5) simplifies and is now expressed by:
(6)$$\sigma_{\psi }=\sqrt{\frac{\sigma_{v_{h}}{}^{2}}{v_{h}{}^{2}}+\sigma_{c}{}^{2}}$$
where $v_{h}$ is the GNSS-derived horizontal velocity (i.e., $v_{h}=\sqrt{v_{e}{}^{2}+v_{n}{}^{2}}$) and $\sigma_{v_{h}}$ is its associated precision. If it is also assumed that the misalignment angle is known (i.e., $\sigma_{c}=0{}^{\circ}$), Equation (6) simplifies and is now expressed by:
(7)$$\sigma_{\psi }=\frac{\sigma_{v_{h}}}{v_{h}}$$

Initializing the heading from a GNSS-derived velocity is only applicable if the forward axis of the navigation reference frame is parallel to the velocity vector, which is a reasonable assumption for most land vehicle navigation applications [21], or if the misalignment angle can be determined. Within UAS-lidar applications, it is common for the forward axis of the navigation reference frame and the GNSS-derived velocity vector to be in different directions. Fixed-wing UAS are affected by crosswind, which can misalign the forward axis and the velocity vector (Figure 4a). Multirotor UAS can move in any direction with respect to the forward axis and therefore can experience misalignment angles of up to 180° (Figure 4b). As a result, the GNSS velocity method can produce incorrect GNSS/INS heading estimates for UAS-lidar applications.

Equation (7) illustrates one of the conceptual relationships between heading precision and GNSS/INS sensor dynamics: as velocity increases, heading precision improves (Figure 4c). However, this statement is not entirely correct as it oversimplifies the complex nature of GNSS/INS integration. In fact, drift from inertial measurement biases will begin to appear in heading estimates when a sensor is moving at a constant velocity (i.e., direction and speed) [13]. The relationship between heading precision and sensor velocity will be further illustrated within the results of the experiment discussed in Section 3.

### 2.3. Effects of Heading Precision on the Positional Precision of a UAS-Lidar Point Cloud

UAS-lidar mapping systems that directly georeference lidar observations require high-accuracy trajectory data to produce high-accuracy point clouds. Understanding how trajectory errors propagate to the positional accuracy of a UAS-lidar point cloud is an important consideration for hardware selection and mission planning. Most GNSS/INS sensor manufacturers report the performance specifications of their products in terms of trajectory errors as shown in Table 2 for the APX-18 performance specifications. Unlike trajectory position errors which directly propagate into positional errors in a point cloud (i.e., coordinate translations), orientation errors propagate into a point cloud in non-linear ways based on the range and direction from the lidar sensor to an observed point. As a result, evaluating the mapping accuracy of a GNSS/INS sensor based on the manufacturer’s specifications can be difficult.

Several published works have rigorously analyzed the effects of trajectory errors on point cloud coordinate accuracies for both UAS-based (e.g., [14]) and occupied aircraft-based laser scanning systems (e.g., [22,23,24]). Thus, a full error analysis is not repeated here; only the effect of heading precision on the positional precision of a point cloud is discussed in order to provide the theoretical foundation for the experiment of this study.

Equation (8) expresses the 3D coordinates of a point (vectors with the subscript *p*) observed by a UAS-lidar system within a north–east–down local level reference frame (superscript *l*). The lidar sensor reference frame is denoted by the superscript *s* and the GNSS/INS sensor reference frame is denoted by the superscript *n*.

(8)$${\left[\begin{array}{c} X\\ Y\\ Z \end{array}\right]}_{p}^{l}=R_{3}\left(\psi \right)R_{2}\left(p\right)R_{1}\left(r\right)\left(R_{s}^{n}\cdot \rho {\left[\begin{array}{c} \sin\left(\alpha \right)\\ \sin\left(\theta \right)\cos\left(\alpha \right)\\ \cos\left(\theta \right)\cos\left(\alpha \right) \end{array}\right]}_{p}^{s}+{\left[\begin{array}{c} l_{x}\\ l_{y}\\ l_{z} \end{array}\right]}^{n}\right)+{\left[\begin{array}{c} X\\ Y\\ Z \end{array}\right]}_{IMU}^{l}$$
where $R_{1}$, $R_{2}$, and $R_{3}$ represent the fundamental rotations around the *x*, *y*, and *z*-axes of a frame, respectively; r,p, and ψ are the roll, pitch, and heading angles, respectively, of the *n* reference frame relative to the *l* frame; $R_{s}^{n}$ represents the boresight rotations from the *s* frame to the *n* frame; ρ, θ, and α are the range, horizontal angle, and vertical angle, respectively, from the lidar sensor to the observed point in the *s* frame; $l_{x}$, $l_{y}$, and $l_{z}$ are the *x*, *y*, *z* lever-arm offsets, respectively, from the center of the IMU to the origin of the lidar sensor measured in the *n* frame; and $X_{IMU}^{l},\;Y_{IMU}^{l},\;\mathrm{and}\;Z_{IMU}^{l}$ are the *X*, *Y*, and *Z* coordinates of the center of the IMU in the *l* frame, respectively.

To isolate the effects of the heading precision, it is assumed that the lidar sensor frame (*s*) and the navigation reference frame (*n*) are perfectly aligned (i.e., $R_{s}^{n}=I$), and that the remaining trajectory estimates (i.e., $r,p,\;X_{nav}^{l},\;Y_{nav}^{l},\;\mathrm{and}\;Z_{nav}^{l}$), calibration parameters (i.e., $l_{x}$, $l_{y}$, and $l_{z}$), and lidar observations (i.e., ρ, θ, α) are all free of error. Using uncorrelated variance propagation, the optimistic coordinate precisions for an observed point in the local level frame are expressed by:(9)$$\sigma_{X_{p}}=\sigma_{\psi }\left|{\left[\begin{array}{c} \sin\left(\psi \right)\cos\left(p\right)\\ \sin\left(\psi \right)\sin\left(p\right)\sin\left(r\right)+\cos\left(\psi \right)\cos\left(r\right)\\ \sin\left(\psi \right)\sin\left(p\right)\cos\left(r\right)-\cos\left(\psi \right)\sin\left(r\right) \end{array}\right]}^{T}\cdot \left[\begin{array}{c} \rho \cdot \sin\left(\alpha \right)+l_{x}\\ \rho \cdot \sin\left(\theta \right)\cos\left(\alpha \right)+l_{y}\\ \rho \cdot \cos\left(\theta \right)\cos\left(\alpha \right)+l_{z} \end{array}\right]\right|$$
(10)$$\sigma_{Y_{p}}=\sigma_{\psi }\left|{\left[\begin{array}{c} \cos\left(\psi \right)\cos\left(p\right)\\ \cos\left(\psi \right)\sin\left(p\right)\sin\left(r\right)-\sin\left(\psi \right)\cos\left(r\right)\\ \cos\left(\psi \right)\sin\left(p\right)\cos\left(r\right)+\sin\left(\psi \right)\sin\left(r\right) \end{array}\right]}^{T}\cdot \left[\begin{array}{c} \rho \cdot \sin\left(\alpha \right)+l_{x}\\ \rho \cdot \sin\left(\theta \right)\cos\left(\alpha \right)+l_{y}\\ \rho \cdot \cos\left(\theta \right)\cos\left(\alpha \right)+l_{z} \end{array}\right]\right|$$
(11)$$\sigma_{Z_{p}}=0$$
where $\sigma_{X_{p}}$, $\sigma_{Y_{p}}$, and $\sigma_{Z_{p}}$ are the precisions in the *X*, *Y*, and *Z* coordinates, respectively, for an observed point within the north–east–down local level frame based only on the precision of the heading estimate. Equation (11) illustrates that heading precision has no effect on the precision of the Z (vertical) coordinate of an observed point. If it is assumed that the UAS-lidar system is operated while nominally level (i.e., r≈0°, p≈0°), Equations (9) and (10) simplify and are now expressed by Equations (12) and (13), respectively.
(12)$$\sigma_{X_{p}}=\sigma_{\psi }\;\left|\sin\left(\psi \right)\left(\rho \cdot \sin\left(\alpha \right)+l_{x}\right)+\cos\left(\psi \right)\left(\rho \cdot \sin\left(\theta \right)\cos\left(\alpha \right)+l_{y}\right)\right|$$
(13)$$\sigma_{Y_{p}}=\sigma_{\psi }\;\left|\cos\left(\psi \right)\left(\rho \cdot \sin\left(\alpha \right)+l_{x}\right)-\sin\left(\psi \right)\left(\rho \cdot \sin\left(\theta \right)\cos\left(\alpha \right)+l_{y}\right)\right|$$

Equations (12) and (13) illustrate that the $l_{z}$ lever-arm offset, under the assumption of nominally level operations, has no effect on the positional precision of an observed point based only on the precision of the heading estimate. As a result, it is commonly recommended to position the origin of the lidar sensor in line with the vertical axis of the navigation reference frame of the GNSS/INS sensor (i.e., $l_{x}=0\;\mathrm{and}\;l_{y}=0$). A similar argument can be made for the alignment of the primary GNSS antenna with respect to the GNSS/INS sensor.

To represent the horizontal precision of an observed point, the distance root mean square measure (dRMS) can be used. Contrary to one-dimensional statistics, there is no fixed probability level for this measure as the probability depends on the ratio of the *X* and *Y* coordinate standard deviations [25]. The dRMS measure is expressed by:(14)$$dRMS_{\psi }=\sqrt{\sigma_{X_{p}}{}^{2}+\sigma_{Y_{p}}{}^{2}}$$
where $dRMS_{\psi }$ is the horizontal precision of an observed point with respect to the precision of the heading estimate. If it is assumed that the $l_{x}$ and $l_{y}$ lever-arm offsets are zero and Equations (12) and (13) are substituted into Equation (14), the $dRMS_{\psi }$ measure is then expressed by:(15)$$dRMS_{\psi }=\sigma_{\psi }\cdot \rho \cdot \sqrt{1-\cos^{2}\left(\alpha \right)\cos^{2}\left(\theta \right)}$$

Equation (15) illustrates the relationship between lidar observations, GNSS/INS heading precision, and the horizontal precision of an observed point under level operating conditions (Figure 5). The heading precision and lidar range observation are the most significant terms within the relationship and are directly proportional to the horizontal precision of an observed point. As a result, reducing the lidar range observations by collecting data from lower UAS flying heights is often employed in practice to mitigate the impact of heading precision.

### 2.4. Single-Antenna versus Dual-Antenna GNSS/INS Heading Precision Experiment

The UAS-lidar mapping system used in this study was an implementation of the open-source Open Mobile Mapping System (OpenMMS) (OpenMMS, Moose Jaw, Canada) [26] and consisted of an Applanix APX-18 UAV GNSS/INS sensor, two Trimble AV14 GNSS antennas (Trimble, Sunnyvale, USA) a Livox Mid-40 lidar sensor (Livox Lidar, Shenzhen, China), and a Sony *α*6000 RGB camera (Sony, Tokyo, Japan) (Figure 6a). The mapping system was installed on a Freefly Systems Alta X UAS (Freefly Systems, Woodinville, USA) (Figure 6b) that operated autonomously during data collection via execution of a pre-planned mission. Raw GNSS and IMU observations were collected by the APX-18 at rates of 5 Hz and 200 Hz, respectively, and the integrated magnetometer was properly calibrated and enabled during the missions. The length of the GNSS compass baseline was measured to be 1.525 m. Lidar observations were collected by the Mid-40 at a rate of 100 kHz within a 38.4° circular field of view (FoV).

The experiment involved collecting UAS-lidar mapping data during four separate flights (mission) as shown in Table 3. Each mission followed the same pre-planned path consisting of nine north–south (N–S) and eight east–west (E–W) flight lines over a project area located near Saskatoon, Saskatchewan, Canada. The project area contained a variety of natural and built features including forested areas, open fields, roofed buildings, and large concrete pads (Figure 7). The UAS-lidar system collected mapping data from a nominal height of 67 m above ground level (AGL) during each mission. However, each mission was flown at a different horizontal velocity in order to examine the effects of sensor dynamics on heading precision (Table 3). At the beginning and end of each flight, an INS initialization procedure was performed following the manufacturer’s instructions [27].

A local GNSS reference station was established at a location within the project area and consisted of a Trimble R10 GNSS receiver/antenna installed on a calibrated tribrach and was secured to a standard wooden-legged tripod. The position of the reference station as shown in Figure 7 was verified to have not changed over the course of the fieldwork. The reference station collected dual-frequency observations from all applicable GNSS satellites based on a 7.5° elevation mask at a rate of 1 Hz for 6.5 h. Upon completion of the fieldwork, the static GNSS observations were uploaded to the online Precise Point Positioning (PPP) service provided by Natural Resources Canada (NRCan) to determine accurate horizontal and vertical datum coordinates for the reference station. While collecting the static observations, the reference station also served as a real-time kinematic (RTK) GNSS base station for a ground survey. The RTK “rover” consisted of a second Trimble R10 GNSS receiver/antenna installed on a calibrated fixed-height survey pole. The RTK survey precisely measured the relative distances (i.e., baselines) from the reference station to 14 ground control points (GCPs) within the project area (Figure 7). Once the datum coordinates for the reference station were determined, the RTK baselines were used to produce 2–3 cm accurate datum coordinates for the GCPs. A localized transverse Mercator projection centered at the reference station with a height-derived scale factor was applied to the GCP horizontal datum coordinates to produce ground-referenced coordinates.

Applanix POSPac UAV software (Applanix, Richmond Hill, Canada) was used to post-process the APX-18 GNSS/INS data with the static reference station GNSS data to produce two separate smoothed best estimate trajectories (SBETs) for each of the four missions. The horizontal and vertical datum parameters and reference station coordinates were input into POSPac. The first processing run had the GAMS functionality enabled. GAMS utilizes the GNSS observations from both antennas in order to leverage the GNSS compass observations within the resulting trajectory estimates. The second processing run had the GAMS functionality disabled with GNSS observations used from only the primary antenna. The SBET position and orientation estimates including the associated precisions were exported to an ASCII file after each processing run. The ASCII trajectory files and the collected lidar data were then processed with OpenMMS software (OpenMMS, Moose Jaw, Canada) to generate two true-color georeferenced point clouds for each mission. The same localized transverse Mercator projection applied to the GCPs was used to produce ground-referenced coordinates for the point clouds. A vertical accuracy assessment was performed on each georeferenced point cloud by comparing the RTK-surveyed GCPs with the nearest lidar point within the point cloud. The assessment followed the American Society for Photogrammetry and Remote Sensing (ASPRS) Positional Accuracy Standards for Digital Geospatial Data [28]. No obvious systematic errors were observed within the point clouds.

The lever-arm offsets between the origin of the lidar sensor and the center of the GNSS/INS sensor were taken from the hardware dimensions within the OpenMMS documentation with a measurement precision of 0.1–0.2 cm [26]. The lever-arm offsets between both the primary and secondary GNSS antennas and the center of the GNSS/INS sensor were determined by a total station survey using angular intersections. This led to measurement precisions of less than 1 cm.

The boresight rotations for both the lidar sensor and the camera with respect to the GNSS/INS sensor’s navigation reference frame were estimated as part of the first mission (i.e., flight A). Specifically, the flight A trajectory file with GAMS enabled was used within the boresight calibration procedures. The estimated boresight rotations were then held constant and applied within the processing of the remaining flights. The lidar sensor boresight calibration utilized a parametric least-squares (PLS) optimization of the RMS errors for planar surfaces found in areas of the point cloud that were observed within overlapping flight lines. Similarly, the camera boresight calibration utilized a PLS optimization of the residuals between the camera’s GNSS/INS-derived exterior orientation parameters (EOPs) and the EOPs resulting from a photogrammetric bundle adjustment. The details of the boresight calibration methods are beyond the scope of this study; see [26] for further information. The generated point clouds were not adjusted by using any additional alignment techniques (e.g., strip adjustment) as they were observed as being well-aligned.

### 2.5. Summary

A primary objective of this study was the assessment of the heading estimates produced by the Applanix APX-18 UAV with and without the use of GAMS. As a commercially available sensor designed for use on multiple UAS platforms, the results of this APX-18 analysis are intended to serve as a representative performance of typical single-antenna and dual-antenna GNSS/INS MEMS-based sensors used within UAS-lidar mapping systems. The methods used to achieve this objective included both theoretical and practical analyses. First, an investigation was conducted into the theoretical concepts underlying the different heading determination techniques employed by GNSS/INS sensors including analytical formulas for the expected precisions. Then, a theoretical analysis was performed examining the impact of heading precision on the positional precision of a UAS-lidar point cloud when utilizing direct georeferencing. Lastly, a field experiment was conducted that collected multiple UAS-lidar mapping datasets over a single project area. Each dataset was collected at a different horizontal speed to ensure the APX-18 experienced a variety of sensor dynamics. The trajectories and point clouds from each dataset were analyzed, compared to expected values and trends, and used to support the conclusions of this study.

## 3. Results

### 3.1. Analysis of Heading Precision Estimates for Flight A

The slow horizontal velocity experienced by the APX-18 sensor during flight A allowed for the best analysis of the impact of using a single versus dual-antenna GNSS/INS sensor to estimate heading. As the horizontal velocity of the UAS was the slowest of all the flights, the differences between the single-antenna and dual-antenna heading precisions would be maximized. Figure 8 illustrates the reported heading precisions during flight A for the single-antenna and dual-antenna trajectories, along with the experienced horizontal velocities and total accelerations. In the figure, note that the software-reported heading precisions based on using a single-antenna and dual-antenna GNSS/INS are shown as the top and bottom plotted lines, respectively, and reference the primary vertical axis. The total accelerations with the horizontal velocities overlaid are shown in the middle of the plot and reference the secondary vertical axis.

As illustrated in Figure 7, each mission began in the north-west corner of the project area and the first flight line was flown in the south direction. Based on this information, Figure 8 indicates that the APX-18 experienced higher accelerations while flying in the south direction, relative to the accelerations experienced while flying in the north direction. The distinguishable ‘spikes’ in the acceleration plot are the result of the UAS quickly starting and abruptly stopping at the beginning and end of each flight line, respectively. It is speculated that higher accelerations experienced while flying south are the result of a 2–3 m/s headwind (see Table 3) causing increased vibrations within the UAS. Although the Alta X UAS utilizes a payload vibration dampening mechanism, increased vibrations were still experienced by the APX-18.

An inspection of Figure 8 indicates varying correlations between the accelerations and the single-antenna and dual-antenna heading precisions. The single-antenna heading precisions for every flight line flown in the south direction are larger at the end of the flight line relative to the start of the flight line, except for the first flight line which was likely collected before the INS had properly initialized. Conversely, the single-antenna heading precisions for the north-flown flight lines are smaller at the end of the flight line relative to the start of the flight line. These results are believed to be attributed to the different accelerations experienced during the north and south-flown flight lines caused by wind. Figure 8 also indicates that the single-antenna heading precisions quickly improved at the end of a N–S flight line (abrupt deceleration) and then quickly worsened at the start of a N–S flight line (quick acceleration followed by zero acceleration). The dual-antenna heading precisions respond in the opposite manner and illustrate a ‘smooth’ response to the experienced accelerations. This is suspected to be a result of the GNSS compass producing a consistent, high-precision heading estimate that is given more observational weight within the proprietary navigation filter. Based on the theoretical GNSS compass model presented in Section 2.2, the expected heading precision for a GNSS compass with a baseline length of 1.525 m is between 0.08° and 0.38° (see Equation (3) and Figure 3), which coincides with the results of flight A.

The heading precisions for the single-antenna GNSS/INS trajectory varied between 0.28° and 0.45° with a mean of 0.369°, and the heading precisions for the dual-antenna GNSS/INS trajectory varied between 0.07° and 0.11° with a mean of 0.082°. These results align with the manufacturer’s heading error performance specifications for the APX-18 sensor shown in Table 2. At slow horizontal velocities, this qualitative analysis suggests that a dual-antenna GNSS/INS sensor significantly improves heading precision by up to a factor of five relative to a single-antenna GNSS/INS sensor.

### 3.2. Analysis of Heading Precision Estimates for All Flights

The different horizontal velocities experienced within the flights allowed for an analysis of the impact of increasing velocity on estimating heading using a single-antenna and dual-antenna GNSS/INS sensor. Figure 9 illustrates the reported heading precisions for the single and dual-antenna GNSS/INS trajectories for all flights. In the figure, note that the legend information indicates the single and dual-antenna GNSS/INS trajectory with a suffix of 1 and 2, respectively. The heading precisions for flight A are shown as dotted lines to emphasize the link between Figure 8 and Figure 9.

An inspection of Figure 9 indicates a negative correlation between horizontal velocity and heading precisions for single-antenna GNSS/INS sensors. The single-antenna heading precisions for flights B, C, and D all improve as the associated horizontal velocities increase. However, the dual-antenna heading precisions are practically unaffected by the increase in horizontal velocity and exhibit a consistent heading precision regardless of the horizontal velocity. Although the single-antenna heading precisions begin to approach the consistent level of precision of the dual-antenna trajectories as horizontal velocity increases (e.g., flight D-1), the lack of GNSS compass heading observations limits the achievable precision of the single-antenna trajectories. Table 4 presents quantitative measures for the single-antenna and dual-antenna heading precisions across the horizontal velocities tested within the experiment. The single-antenna heading precisions of flight B improve in response to the experienced accelerations, similar to the relationship of the single-antenna heading precisions shown in Figure 8. Conversely, the single-antenna heading precisions of flight C and D worsened in response to the accelerations, similar to the relationship of the dual-antenna heading precisions shown in Figure 8. This suggests that a horizontal velocity threshold may exist between 5 m/s and 10 m/s, in which the beneficial influences of the accelerations on heading precision become a detrimental influence.

For each flight, the mean horizontal velocity of the APX-18 during the longer north–south flight lines was calculated. The shorter east–west flight lines were excluded as the APX-18 was moving at a considerably lower velocity during these flight lines, which introduced a bias within the mean velocity measures. The mean horizontal velocities and associated mean single-antenna heading precisions were then considered as observations produced by a theoretical GNSS velocity model (see Section 2.2, Equation (6)) and were used to estimate the model’s GNSS-derived horizontal velocity precision (*σ_vh_*) and heading precision achievable limit (*σ_c_*) parameters via a general least-squares adjustment. The resulting estimates were 0.015 m/s for *σ_vh_*, which agrees with the manufacturer’s velocity error performance specification for the APX-18 sensor shown in Table 2, and 0.148° for *σ_c_*. These results suggest that the GNSS velocity model may be a reasonable estimator of the expected heading precision for a single-antenna GNSS/INS sensor based on horizontal velocity. They also suggest that a point of diminishing returns for the improvement of heading precisions exists at a horizontal velocity of approximately 15 m/s for single-antenna GNSS/INS sensors. Figure 10 illustrates the GNSS velocity model that best fits the mean velocity and single-antenna heading precision estimates from the experiment.

### 3.3. Analysis of the Remaining Trajectory Precision Estimates for All Flights

Processing the GNSS/INS data with and without the use of GAMS had a negligible impact on the mean precisions of the remaining single and dual-antenna GNSS/INS trajectory estimates as shown by the difference in means, column (Δ) in Table 5. This table presents quantitative measures for all the single and dual-antenna trajectory precisions across the horizontal velocities tested within the experiment. The heading precision measures as presented in Table 4 are included for completeness.

### 3.4. Analysis of the Point Cloud Horizontal and Vertical Positional Precision for Flight A

The slow horizontal velocity experienced by the APX-18 sensor during flight A allowed for the best analysis of the impact of heading precision on the positional precision of a UAS-lidar point cloud. As the horizontal velocity of the UAS was the slowest of all flights, the APX-18 sensor was closest to being in a level orientation during this mission. The total tilt angle of the sensor, calculated with respect to the nadir direction, was typically less than 10° during each N–S flight line.

A point observed near the edge of the circular field of view (FoV) of the Mid-40 lidar sensor is most affected by the precision of the trajectory heading estimate. The observable horizontal angle (*θ*) and vertical angle (*α*) for a point at the edge of a circular FoV are related by cos(*θ*)cos(*α*) = cos(W/2), where W is the total FoV angle (e.g., W = 38.4° for the Mid-40). As a result, the two georeferenced point clouds produced from the single-antenna and dual-antenna trajectories for flight A were filtered to only include points that were observed within one of four narrow edge regions of the Mid-40′s FoV. For comparison, an additional narrow region of points least affected by the precision of the heading estimate, located at the middle of the FoV, were also included for analysis (Table 6 and Figure 11 and Figure 12). Each point within the single-antenna generated point cloud was matched to its corresponding point within the dual-antenna generated point cloud and the horizontal and vertical distances between the matched points were calculated. Figure 13 illustrates a north–west looking-perspective view of the point cloud for flight A, colored by gray-scale intensity values. The analysis presented here focuses on relative differences, although a higher accuracy reference point cloud (e.g., produced from a terrestrial laser scanner) may allow an absolute horizontal accuracy assessment of each respective point cloud and will thus be the focus of future research. The calculated positional differences between the matched points were used to estimate the impact of the precision of the heading estimate on the positional precision of a UAS-lidar point cloud to compare with theoretical expectations.

Figure 14 illustrates the horizontal position differences between the single and dual-antenna generated point clouds based on the points within the five regions of the Mid-40′s FoV. In the figure, note that the heading precisions for the single-antenna GNSS/INS trajectory are shown as the top dashed line and reference the secondary vertical axis.

An inspection of Figure 14 indicates varying correlations between the horizontal position differences, the single-antenna heading precisions, and the direction each flight line was flown. Excluding the first flight line, every flight line flown in the south direction unexpectedly yielded smaller horizontal position differences than flight lines flown in the north direction. Table 7 presents quantitative measures for the horizontal position differences between the point clouds within the five FoV regions for flight A.

Further analysis of the north (Y) and east (X) coordinate differences of the horizontal position differences within the four edge regions suggested the presence of a bias within the heading estimates for the single-antenna GNSS/INS trajectory, relative to the dual-antenna GNSS/INS trajectory. Computing the differences between the heading estimates from the single and dual-antenna GNSS/INS trajectories produced a mean difference of 0.27°. To investigate if this heading bias contributed to the horizontal position differences being considerably different for the north and south flow flight lines, it was used to adjust the single-antenna GNSS/INS heading estimates. Then, the horizontal position differences between the matched points of the single-antenna and dual-antenna point clouds were recomputed. Figure 15 illustrates the horizontal position differences between the single and dual-antenna generated point clouds after adjusting for the heading bias.

An inspection of Figure 15 indicates that the horizontal position differences have been reduced, are more consistent, and are less correlated with the direction each flight line was flown after adjusting for the single-antenna GNSS/INS trajectory heading bias. This suggests that the single-antenna GNSS/INS heading estimates reported by the APX-18 could potentially be improved through data-driven calibrations and this will be the focus of future research.

Using the theoretical point cloud horizontal precision model presented in Section 2.3 (see Equation (15)), an estimate for the horizontal precision of a point observed at the edge of the Mid-40′s circular FoV was computed. Based on the mean single-antenna heading precision of 0.369°, the horizontal precision for an edge FoV point observed at a lidar range of 68 m was 0.15 m, which closely aligns with the maximum horizontal position differences computed after adjusting for the heading bias (Table 8).

Equation (15) also indicates that the horizontal position of a point at the middle of the FoV is not affected by the precision of the heading estimate. Therefore, the expected horizontal position difference between the two point cloud regions would be zero. However, Figure 15 and Table 8 indicate a mean horizontal position difference for the points in the middle region of 0.043 m. This is the suspected result of not accounting for the remaining trajectory estimates, calibration parameters, and lidar observations containing errors (i.e., the assumption that these terms are error free was used within the development of Equation (15), which is invalid).

Figure 16 illustrates the vertical position differences between the single and dual-antenna generated point clouds based on the points located within the five regions of the Mid-40′s FoV. In the figure, note that the heading precisions for the single-antenna GNSS/INS trajectory are shown as the top dashed line and reference the secondary vertical axis.

Equation (11) indicates that heading precision has no effect on the vertical coordinate of an observed lidar point. Therefore, the vertical position differences between the single and dual-antenna point clouds within any of the five regions would be zero. An inspection of Figure 16 indicates that the vertical position differences for points within the left, right, and middle regions are approximately zero, but the points within the top and bottom regions have opposite mean vertical position differences of 0.015 m and −0.013 m, respectively. As the top and bottom regions closely align with the direction of travel of the Mid-40 lidar sensor as shown in Figure 11, the vertical position differences are suspected to be a result of a small relative bias in the single-antenna GNSS/INS trajectory pitch estimates compared to the dual-antenna solution. Table 9 presents quantitative measures for the vertical position differences between the point clouds within the five FoV regions for flight A.

## 4. Discussion and Conclusions

Due to the wide adoption of smaller, lower-cost MEMS-based IMUs within GNSS/INS solutions, hardware sensors that are suitable for UAS applications now exist. However, the ability to accurately determine heading with such sensors is often challenging, especially at the lower velocities often experienced during UAS operations. By utilizing two GNSS antennas, a real-time differential GNSS solution leveraging the millimeter precise carrier phase observable to determine heading is possible. Though widely employed in ground-based and sea-based applications, dual-antenna GNSS/INS sensors are not commonly used onboard UAS due to the increased cost, weight, and complexity of installing the dual GNSS antennas with an adequate separation onboard the airframe of the UAS. The principal objective of this study was to assess and understand the impacts of using either a single-antenna or a dual-antenna GNSS/INS sensor for UAS-lidar applications by collecting and analyzing field-collected data from a typical UAS-lidar mapping system.

The positioning and navigation principles involved within GNSS and inertial-based systems were presented with a specific focus on heading determination. The theoretical relationship between a GNSS compass and heading precision was discussed, for which the results of this study supported the theory and provided compelling arguments that heading precision can be improved by up to a factor of five when a dual-antenna GNSS/INS sensor is used. Additionally, the theoretical relationship between the horizontal velocity of a GNSS/INS sensor and the heading precision was discussed and validated through the results of the four UAS-lidar flights conducted within this study. At higher horizontal velocities, a single-antenna GNSS/INS sensor can estimate heading to a level of precision that approaches that of a dual-antenna GNSS/INS sensor, but even at 15 m/s, the single-antenna heading estimates are still less precise than the dual-antenna estimates by a factor of 1.5. Additionally, a downside of collecting lidar observations while flying at higher horizontal velocities concerns a decrease in the point density of the generated point cloud. Interpolating between the experimental results of the test flights, an argument can be made that a balance between improving heading precision and maintaining point cloud density exists at a horizontal velocity between 5 m/s and 10 m/s for single-antenna GNSS/INS sensors. Dual-antenna GNSS/INS sensors are practically unaffected by horizontal velocity and maintain the ability to provide higher-precision heading estimates given adequate satellite visibility and a low multipath environment. The results of this study also indicated that the precisions of the remaining roll, pitch, and positional GNSS/INS trajectory estimates were negligibly impacted by using a single-antenna versus a dual-antenna or by changes in the horizontal velocity. Lastly, the theoretical relationship between the experienced heading precision of a GNSS/INS sensor and the positional precision of a resulting point cloud was discussed, for which the results of this study supported the theory and provided compelling arguments that a dual-antenna GNSS/INS sensor consistently produces a point cloud with improved 3D positional precision.

In addition to the precision assessments performed in this study, discussions were presented that aim to provide useful information to UAS-lidar mapping practitioners with respect to sensor placement and alignment, the importance of sensor dynamics (especially for single-antenna GNSS/INS sensors), and mission planning considerations (e.g., flying height, flying speed, and the impacts of a lidar sensor’s FoV). As reported in existing literature and confirmed by this study, the precision of the heading estimates of a GNSS/INS sensor directly impacts the 3D positional precision of a point cloud produced from a UAS-lidar mapping system, notably more so than any other trajectory estimate. Although the use of dual-antenna GNSS/INS sensors can increase the cost and complexity of a UAS-lidar mapping system, investing in a system with a dual-antenna GNSS/INS sensor is recommended, as it provides a simple solution to realizing higher-precision heading estimates and as a result, more accurate and precise point clouds.

## Figures and Tables

**Figure 1 sensors-21-05382-f001:**
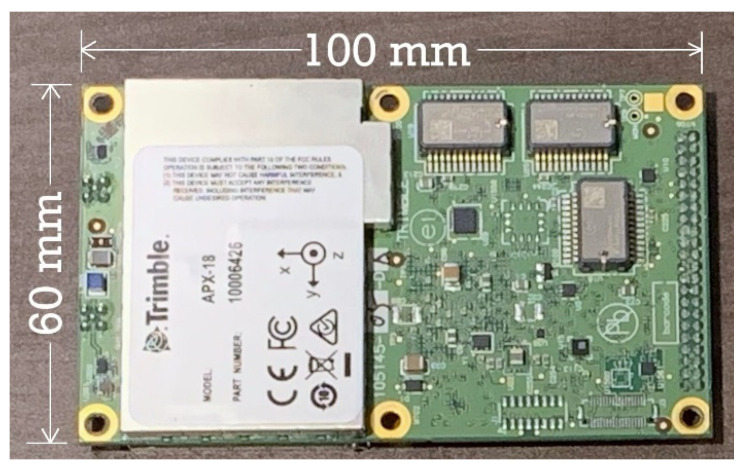
Applanix APX-18 UAV dual-antenna GNSS/INS sensor.

**Figure 2 sensors-21-05382-f002:**
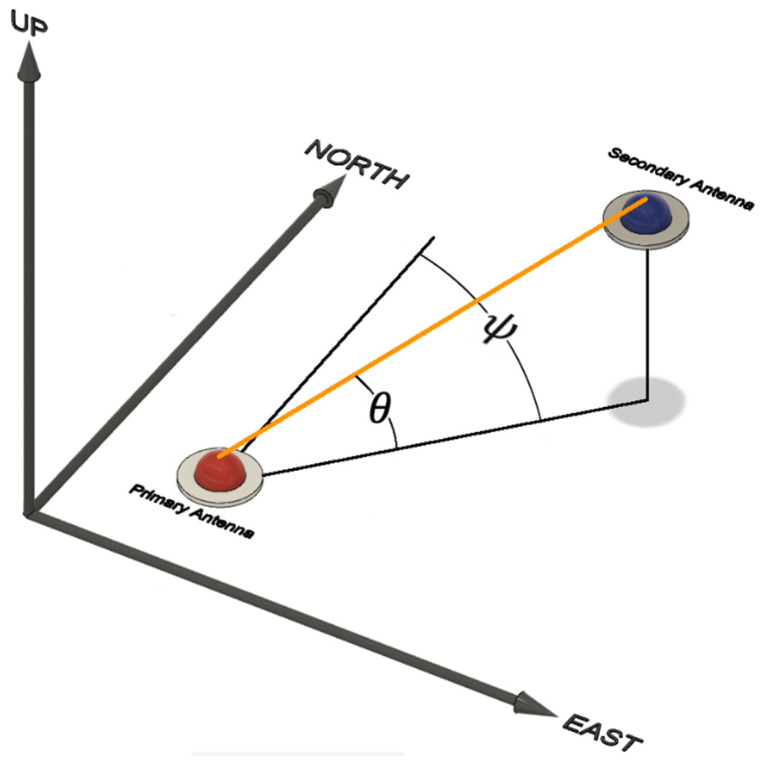
GNSS compass heading (*ψ*) and tilt (*θ*) angle estimates expressed within a local level (east–north–up) frame.

**Figure 3 sensors-21-05382-f003:**
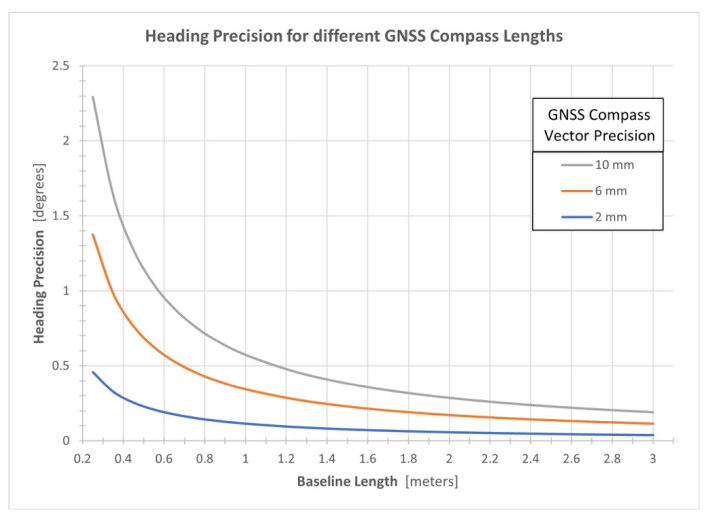
Theoretical GNSS compass heading precisions for different GNSS compass baseline lengths.

**Figure 4 sensors-21-05382-f004:**
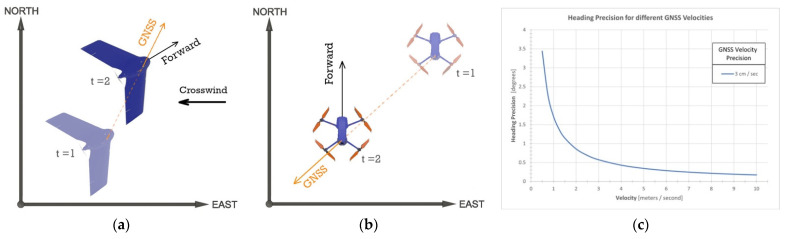
Misalignment of the forward axis of the navigation reference frame and the GNSS velocity vector for (**a**) fixed-wing UAS and (**b**) multirotor UAS. (**c**) Theoretical heading precision as a function of GNSS-derived velocity.

**Figure 5 sensors-21-05382-f005:**
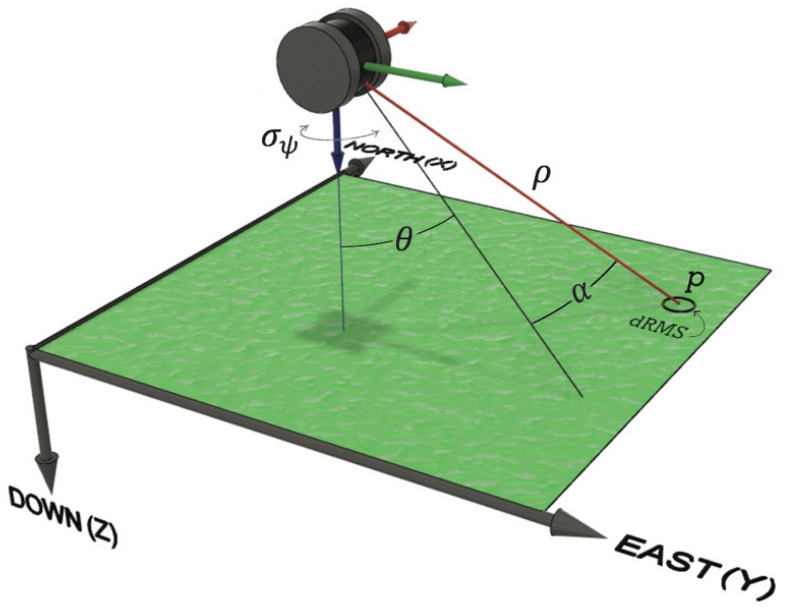
A local level (north–east–down) frame illustrating an observed ground point, the lidar observations, the heading precision, and the horizontal precision of the ground point.

**Figure 6 sensors-21-05382-f006:**
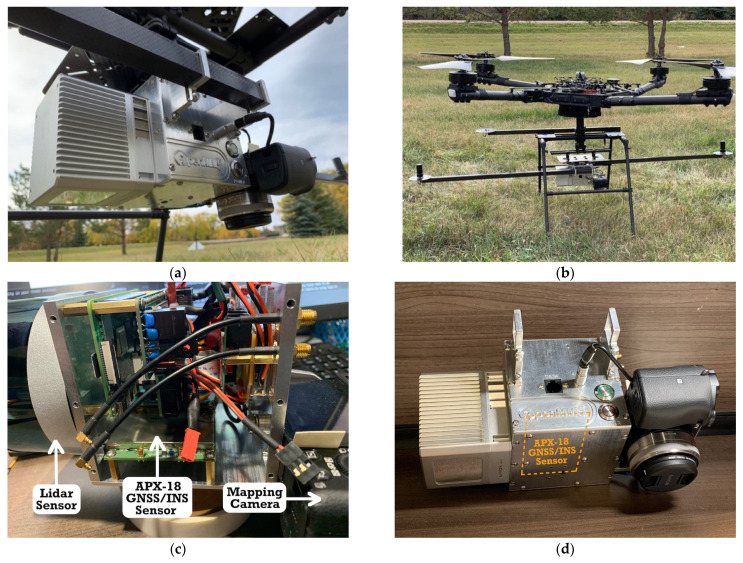
(**a**) OpenMMS UAS-lidar mapping system used in this study. (**b**) Mapping system installed on a Freefly Systems Alta X UAS. (**c**) Internal configuration of the APX-18 GNSS/INS sensor. (**d**) Relative positions of the mapping sensors.

**Figure 7 sensors-21-05382-f007:**
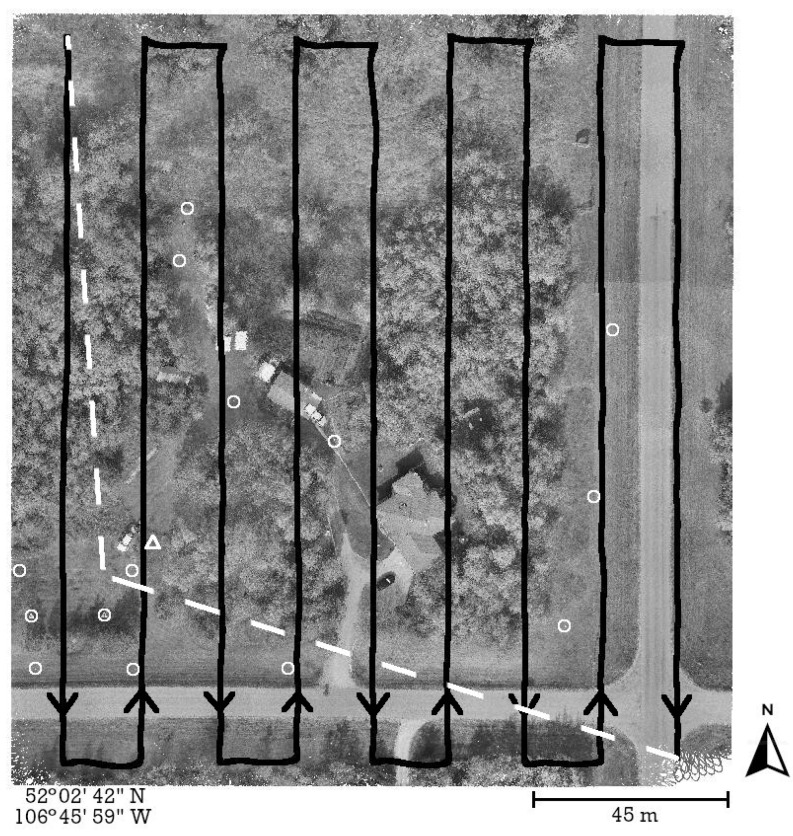
Project area located near Saskatoon, Saskatchewan. INS initialization segments are shown as white dashed lines and mission flight lines are shown in black. The GNSS reference station is shown as a triangle and the GCPs as circles.

**Figure 8 sensors-21-05382-f008:**
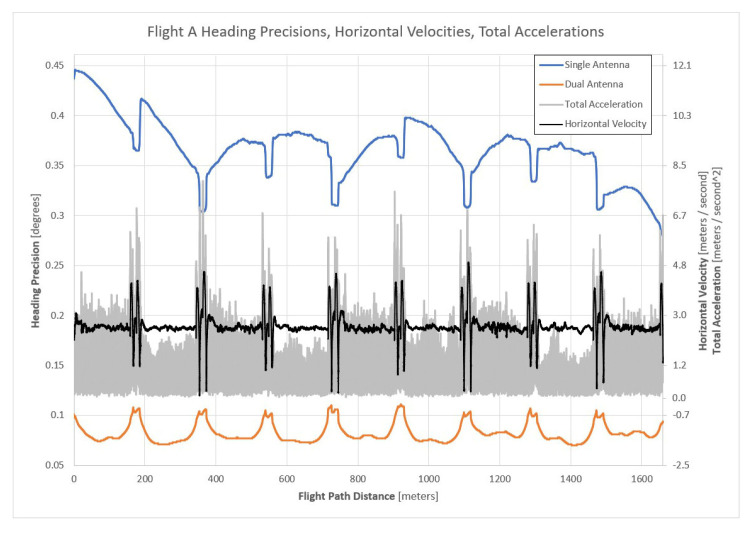
Single and dual-antenna GNSS/INS heading precisions, horizontal velocities, and total accelerations for flight A of the experiment.

**Figure 9 sensors-21-05382-f009:**
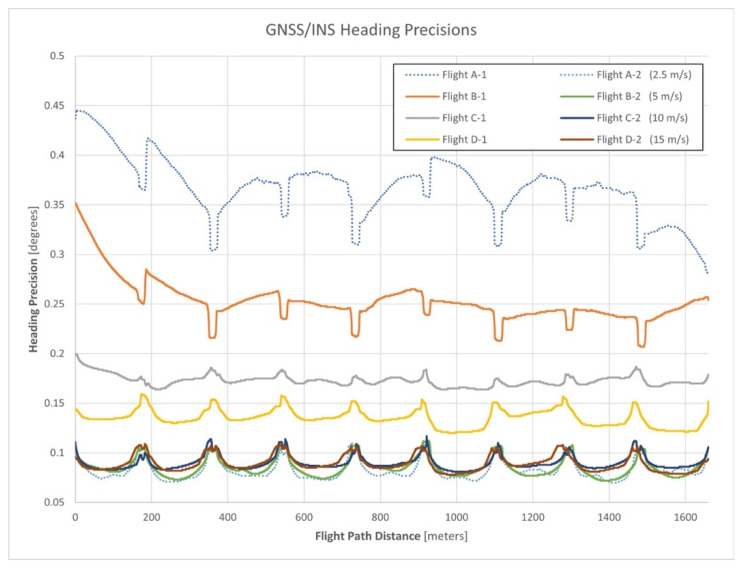
Single and dual-antenna GNSS/INS heading precisions for all flights of the experiment.

**Figure 10 sensors-21-05382-f010:**
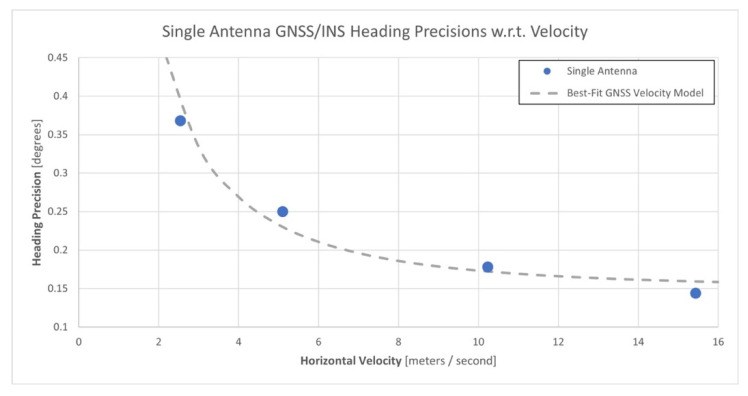
Single-antenna GNSS/INS heading precisions at the different horizontal velocities of the experiment best-fit to the theoretical GNSS velocity model.

**Figure 11 sensors-21-05382-f011:**
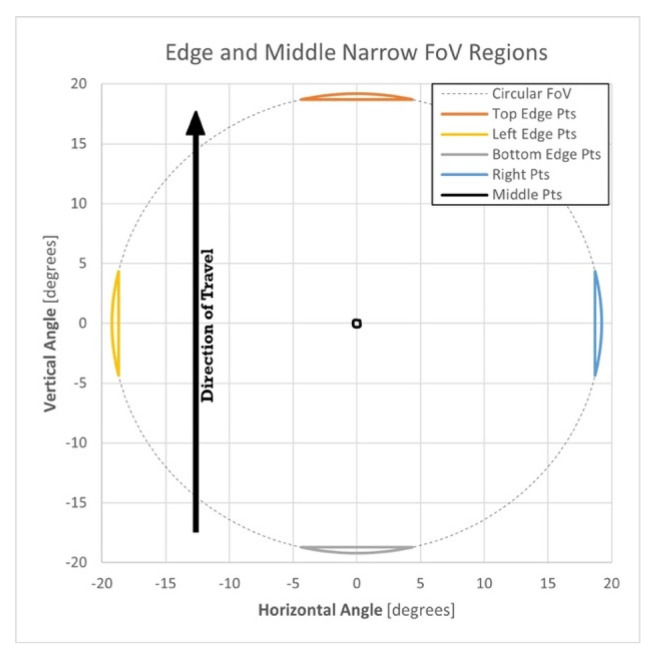
Five regions in the circular field of view of the Mid-40 lidar sensor used for analysis.

**Figure 12 sensors-21-05382-f012:**
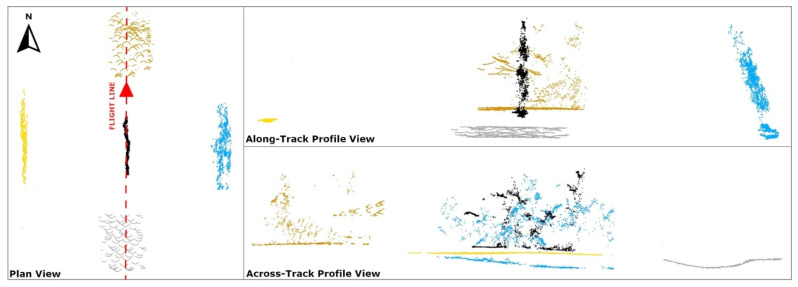
Views of a point cloud containing 5 s of collected data from the five FoV regions of the Mid-40 sensor.

**Figure 13 sensors-21-05382-f013:**
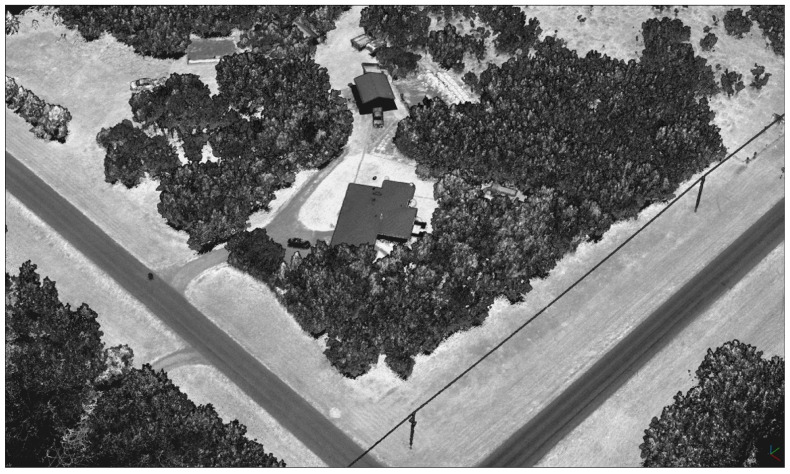
North–west looking-perspective view of the point cloud for flight A, colored by gray-scale intensity values.

**Figure 14 sensors-21-05382-f014:**
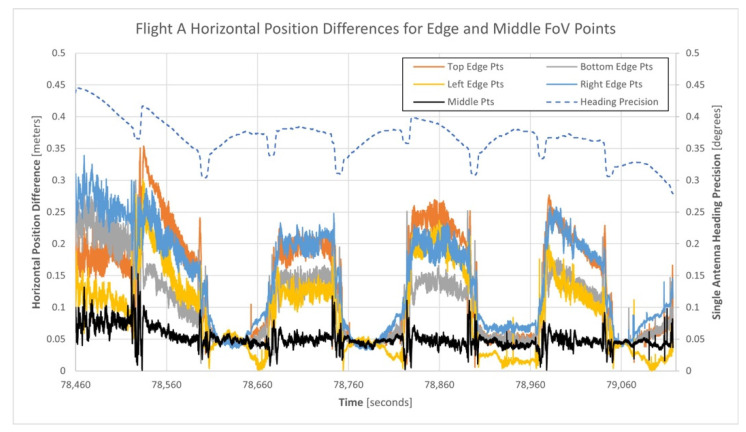
Horizontal position differences between the single and dual-antenna generated point clouds for the five regions of points of flight A.

**Figure 15 sensors-21-05382-f015:**
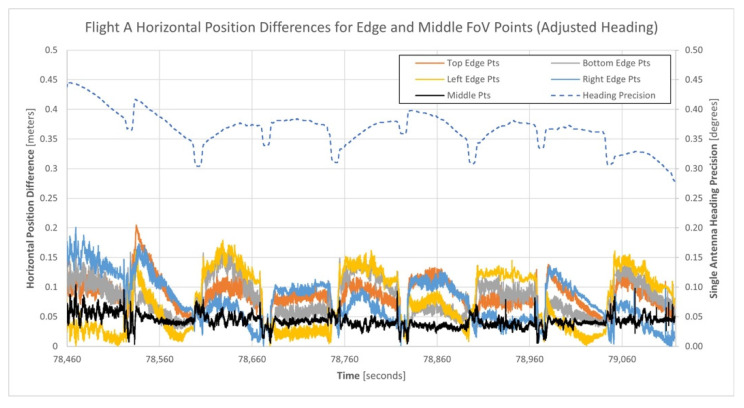
Horizontal position differences between the single and dual-antenna generated point clouds for the five regions of points of flight A after adjusting for the heading bias.

**Figure 16 sensors-21-05382-f016:**
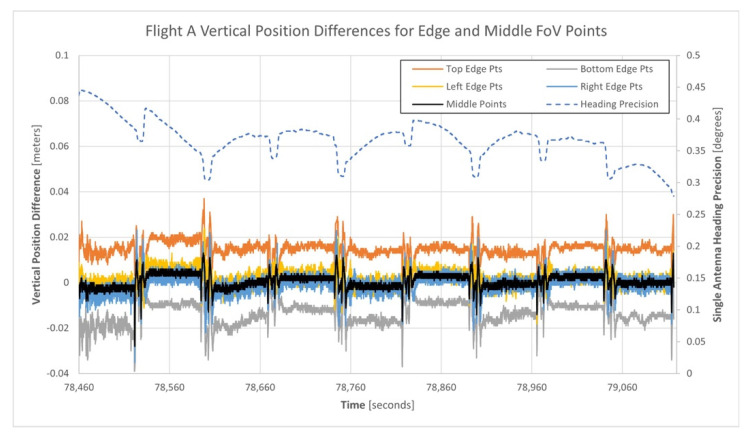
Vertical position differences between the single and dual-antenna generated point clouds for the five regions of points of flight A.

**Table 1 sensors-21-05382-t001:** Summary of GNSS and INS characteristics (Source: [9]).

Characteristic	GNSS	INS
Accuracy of navigation solution	High long-term accuracy but noisy in short-term	High short-term accuracy but deteriorates with time
Initial conditions	Not required	Required
Orientation information	Typically not available ^1^	Available
Sensitive to gravity	No	Yes
Self-contained	No	Yes
Jamming immunity	No	Yes
Output data rate	Low	High

^1^ With multiple antennas, some GNSS receivers can render orientation information as well.

**Table 2 sensors-21-05382-t002:** Applanix APX-18 UAV performance specifications and root mean square error (Source: [16]).

	SPS ^1^	RTK	PP-RTX ^2^	Post-Processed
Position (m)	1.5–3.0	0.02–0.05	0.03–0.06	0.02–0.05
Velocity (m/s)	0.05	0.02	0.015	0.015
Roll and Pitch (deg)	0.04	0.03	0.025	0.025
True Heading (deg)	0.15	0.10	0.08	0.080

^1^ Standard positioning service (i.e., autonomous GNSS positioning). ^2^ Post-processed CenterPoint^®^ RTX™ service.

**Table 3 sensors-21-05382-t003:** Characteristics of each mission.

Flight Name	Horizontal Velocity	Duration ^1^	Wind Direction (from)	Wind Speeds
A	2.5 m/s	657 s	S	2–3 m/s
B	5 m/s	351 s	S–SW	1–2 m/s
C	10 m/s	227 s	S	2–3 m/s
D	15 m/s	178 s	S	2–3 m/s

^1^ From the start of the first flight line to the end of the last flight line.

**Table 4 sensors-21-05382-t004:** Quantitative measures for single and dual-antenna GNSS/INS heading precisions.

Flight	Mean Horizontal Velocity *	Single-Antenna Heading Precisions			Dual-Antenna Heading Precisions			Improvement in Heading Precision
		Min.	Max.	Mean *	Min.	Max.	Mean	
A	2.5 m/s	0.28°	0.45°	0.369°	0.07°	0.11°	0.082°	4.5×
B	5.1 m/s	0.21°	0.35°	0.251°	0.07°	0.11°	0.086°	2.9×
C	10.2 m/s	0.16°	0.20°	0.175°	0.08°	0.12°	0.093°	1.9×
D	15.4 m/s	0.12°	0.16°	0.141°	0.08°	0.11°	0.094°	1.5×

* Observations used within the GNSS velocity model analysis as discussed below.

**Table 5 sensors-21-05382-t005:** Quantitative measures for single-antenna and dual-antenna GNSS/INS trajectory precisions.

Flight	Hor. Velocity	Trajectory Estimates	Single-Antenna Precisions			Dual-Antenna Precisions			Δ
			Min.	Max.	Mean	Min.	Max.	Mean	
A	2.5 m/s	North	0.019 m	0.025 m	0.020 m	0.019 m	0.023 m	0.020 m	0.000 m
		East	0.019 m	0.024 m	0.020 m	0.018 m	0.022 m	0.019 m	0.001 m
		Height	0.036 m	0.036 m	0.036 m	0.036 m	0.036 m	0.036 m	0.000 m
		Roll	0.03°	0.07°	0.038°	0.03°	0.06°	0.037°	0.001°
		Pitch	0.03°	0.07°	0.039°	0.03°	0.06°	0.037°	0.002°
		Heading	0.28°	0.45°	0.369°	0.07°	0.11°	0.082°	0.287°
B	5 m/s	North	0.019 m	0.024 m	0.020 m	0.019 m	0.023 m	0.020 m	0.000 m
		East	0.019 m	0.022 m	0.020 m	0.018 m	0.022 m	0.020 m	0.000 m
		Height	0.036 m	0.036 m	0.036 m	0.036 m	0.036 m	0.036 m	0.000 m
		Roll	0.03°	0.07°	0.039°	0.03°	0.06°	0.038°	0.001°
		Pitch	0.03°	0.07°	0.041°	0.03°	0.06°	0.039°	0.003°
		Heading	0.21°	0.35°	0.251°	0.07°	0.11°	0.086°	0.165°
C	10 m/s	North	0.019 m	0.022 m	0.021 m	0.019 m	0.022 m	0.021 m	0.000 m
		East	0.019 m	0.024 m	0.021 m	0.019 m	0.022 m	0.021 m	0.000 m
		Height	0.036 m	0.037 m	0.036 m	0.036 m	0.037 m	0.036 m	0.000 m
		Roll	0.03°	0.08°	0.045°	0.03°	0.08°	0.042°	0.003°
		Pitch	0.03°	0.07°	0.041°	0.03°	0.07°	0.041°	0.000°
		Heading	0.16°	0.20°	0.175°	0.08°	0.12°	0.093°	0.082°
D	15 m/s	North	0.020 m	0.022 m	0.021 m	0.020 m	0.022 m	0.021 m	0.000 m
		East	0.020 m	0.024 m	0.022 m	0.020 m	0.023 m	0.021 m	0.001 m
		Height	0.035 m	0.038 m	0.036 m	0.035 m	0.038 m	0.036 m	0.000 m
		Roll	0.03°	0.09°	0.051°	0.03°	0.08°	0.048°	0.003°
		Pitch	0.03°	0.07°	0.046°	0.03°	0.07°	0.045°	0.001°
		Heading	0.12°	0.16°	0.141°	0.08°	0.11°	0.094°	0.047°

**Table 6 sensors-21-05382-t006:** Regions in the circular field of view of the Mid-40 lidar sensor used for analysis.

Region Name	Minimum Hor. Angle	Maximum Hor. Angle	Minimum Vert. Angle	Maximum Vert. Angle
Top Edge Points	−4.4° *	4.4° *	18.7°	19.2°
Bottom Edge Points	−4.4° *	4.4° *	−19.2°	−18.7°
Left Edge Points	18.7°	19.2°	−4.4° *	4.4° *
Right Edge Points	−19.2°	−18.7°	−4.4° *	4.4° *
Middle Points	−0.25°	0.25°	−0.25°	0.25°

* Calculated value shown only for completeness and not used to filter the point clouds.

**Table 7 sensors-21-05382-t007:** Quantitative measures for the horizontal position differences of points within the five regions of interest for flight A.

Region	Horizontal Position Differences			Mean Lidar Range	No. of Points
	Minimum	Maximum	Mean		
Top Edge Points	0.006 m	0.354 m	0.137 m	68.2 m	263,266
Bottom Edge Points	0.012 m	0.300 m	0.105 m	68.8 m	258,259
Left Edge Points	0.000 m	0.302 m	0.092 m	68.1 m	285,623
Right Edge Points	0.033 m	0.339 m	0.145 m	68.2 m	286,220
Middle Points	0.001 m	0.164 m	0.051 m	64.6 m	297,850

**Table 8 sensors-21-05382-t008:** Quantitative measures for the horizontal position differences of points within the five regions of interest for flight A after adjusting for the heading bias.

Region	Horizontal Position Differences		
	Minimum	Maximum	Mean
Top Edge Points	0.009 m	0.205 m	0.087 m
Bottom Edge Points	0.004 m	0.168 m	0.080 m
Left Edge Points	0.001 m	0.179 m	0.075 m
Right Edge Points	0.001 m	0.201 m	0.078 m
Middle Points	0.004 m	0.104 m	0.043 m

**Table 9 sensors-21-05382-t009:** Quantitative measures for the vertical position differences of points within the five regions of interest for flight A.

Region	Vertical Position Differences		
	Minimum	Maximum	Mean
Top Edge Points	−0.004 m	0.037 m	0.015 m
Bottom Edge Points	−0.039 m	0.005 m	−0.013 m
Left Edge Points	−0.034 m	0.025 m	0.002 m
Right Edge Points	−0.035 m	0.023 m	−0.001 m
Middle Points	−0.028m	0.013 m	0.001 m

## Data Availability

Not applicable.

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
