# Peer review of "Investigating Practical Impacts of Using Single-Antenna and Dual-Antenna GNSS/INS Sensors in UAS-Lidar Applications"

_sensors, 2021, doi:10.3390/s21165382_

Round 1

Reviewer 1 Report

The authors of the manuscript take on a very important issue. The paper is written with care. Taking into account the nature of the work, which describes the results of the conducted experiment, it should first of all be supplemented with a discussion of how the obtained results relate to the results of other researchers. Only after such an analysis will the manuscript be fully valuable. The introduction of the article should be supplemented with other literature to provide sufficient background and include all relevant references.

Chapter 2 of the manuscript is very extensive and contains many obvious formulas and definitions. This hardly emphasises the authors' contribution to the existing state of knowledge. The structure of the chapter should be reformatted. Unfortunately, the name appears in subtitle 2.2 Summary of Methods. Subsequently, the following subchapters are described. The summary should usually appear at the end of the chapter.

The results of the experiment are described sufficiently.

The chapter on discussion and conclusions should be restructured. Some of the conclusions are obvious and very general (e.g. lines 717-727) and fit more into the introduction.

Author Response

An additional reference has been added to the introduction to help provide sufficient background. If there are any specific areas within the introduction that require additional background or mention to specific references the reviewer is aware of, the authors would gladly revise the manuscript accordingly.

The authors are unaware of extensive research performed by other researchers that could be compared to the results presented, besides the research and results already discussed within the manuscript. Any specific literature that could be used to increase the value of the manuscript would gladly be included.

The authors agree that some of the formulas and definitions are trivial, but they have to carefully selected for inclusion within the manuscript in order to provide the necessary background theory so that a wider range of researchers and practitioners may find value in this research.

The previous 'Summary of Methods' section has been renamed to 'Summary' and has been moved to the end of Chapter 2. The authors thank the reviewer for this recommendation.

The introduction and conclusion sections have been restructured and hopefully provide an appropriate amount of detail and commonality. Specific quantitative metrics have been reiterated within the discussion and conclusions section so to provide a reader with the necessary findings of the manuscripts in a succinct manner.

Reviewer 2 Report

The paper "Investigating practical impacts of using single-antenna and dual-antenna GNSS/INS sensors in UAS-lidar applications" is in general well written and aimes at quantifying the performance of estimating the heading using a single-antenna GNSS receiver versus a dual-antenna GNSS receiver with typical UAS operations. The topic is timely and clearly demonstrates the performance of these two types of receivers through dedicated experiments with a drone.

Although interesting from a practical point of view, the scientific contribution from the presented methodology is somewhat low.

ln. 17: The argument that increased complexity is one of the downsides of obtaining a dual-antenna GNSS receivers seems somewhat vague.

ln. 17-19: The problem formulation could have been be more clear. The term "uncertainty" as a quality indicator needs to be properly defined.

ln. 86: GNSS is a part of an INS, thus it is the observations from GNSS and IMUs that are fused with e.g. EKF.

Figure 2: The coordinate frame presented is NEU, thus being a left hand system. I suggest using NED frame instead.

Figure 2 / Equation 2a: There is a mismatch between the figure and the sine/cosine coefficients in the corresponding equation. They seem to have been swapped.

Equation 5: The coordinate frames related to the chosen indexes needs to be defined, i.e. s-sensor frame, n-navigation frame, etc.. It should be easy for the reader to follow all the frame transitions in the equation from the indexes - this is not the case now.

Figure 8: The zero-level on the right axis (i.e. velocity/acceleration) should be ticked.

ln. 544: ...dual-antenna GNSS/INS sensor significantly reduces heading uncertainty...

Figure 9: Explain more clearly why there still is a bias in the heading uncertainty between Flight D-1 and Flights A2-D2 even at 15 m/s? 

Author Response

The authors agree that the pure scientific contributions of this research may be somewhat low, however a singular analysis of the type presented within the manuscript does not exist within the current body of knowledge, and as such, the authors feel the research does offer value to both academic researchers and industry practitioners. 

The authors thank the reviewer for identifying the vagueness of the 'complexity' involved within a dual-antenna sensor. This point has now been further explained within the manuscript.

The fairly generalized term of 'uncertainty' has now been replaced by the more meaningful term of 'precision' throughout the manuscript. The authors thank the reviewer for this valuable recommendation.

The fusion of GNSS and IMU observations has now been clarified within the manuscript.

The local level coordinate system in Figure 2 was meant to be a right-handed East-North-Up frame, this has now been clarified within the manuscript.

The authors have reviewed Equation 2a with respect to the reference it is from as well as from an independent 'from scratch' analysis. The Equation as presented is correct.

The authors have attempted to better explain the coordinate frames (and indexes) involved within Equation 5 before the equation is presented within the manuscript and therefore aid in a readers ability to follow along within the transformation operations being performed.

The zero axis and label have now been added to Figure 8. The authors thank the reviewer for her/his attention to detail with this recommendation.

On the line 544. the word 'significantly' has now been added to the manuscript as recommended.

The heading bias between the fastest single-antenna flight heading estimates and the dual-antenna heading estimates has now been further explained within the manuscript.

Round 2

Reviewer 1 Report

I thank the authors for the changes they have made and the clarifications they have provided. I will once again emphasize the practical nature of the research done. I have one more comment to consider for the authors. The manuscript could be supplemented with a visualization of the location of the GNSS receiver on the unmanned aerial vehicle (e.g. extension of Figure 1 or 6).

Author Response

Thank you for the recommendation, figure 6 has been updated to include two additional images to illustrate the installation of the APX-18 GNSS/INS sensor within the MMS, and the relative positions of all the mapping sensors.